# Would You Detour with Me? Association between Functional Breed Selection and Social Learning in Dogs Sheds Light on Elements of Dog–Human Cooperation

**DOI:** 10.3390/ani13122001

**Published:** 2023-06-15

**Authors:** Petra Dobos, Péter Pongrácz

**Affiliations:** Department of Ethology, ELTE Eötvös Loránd University, Pázmány Péter sétány 1/c, 1117 Budapest, Hungary; dobospetra1@gmail.com

**Keywords:** dog, social learning, human demonstrator, independent breeds, cooperative breeds, functional selection

## Abstract

**Simple Summary:**

Dogs have become inseparable from humans and show an excellent capacity to coexist and cooperate with us. Currently, there is vivid interest among scientists and enthusiasts towards the origins of those skills that enable dogs to embrace the perfect companion role. Many wish to know whether particular dog breeds, or breed types, show differences when it comes to interacting with humans. Here, we used a robust grouping criterion—whether a dog belongs to the cooperative or independent working dog type—to see if work-related selection in the past may affect how these breeds learn from the behaviour of a human. In this study, dogs had to obtain a treat/toy reward from behind a V-shaped fence by performing a detour in three consecutive trials. Our main result showed that cooperative dogs improved their detour speed when they observed the demonstrator; however, independent dogs did not improve. As we also included several non-related breeds in the groups, these results indicate that selecting for cooperativity or individual working ability in various dog breeds could affect dogs’ capacity and willingness to pay close attention to the behaviour of humans. Our results provide an important insight into the understanding of how dogs match their behaviour to ours in everyday situations.

**Abstract:**

Interspecific social learning is a main synchronizing mechanism that enables dogs to adapt to the anthropogenic niche. It is not known whether dogs in general possess the capacity of learning from humans or whether more recent selective events have affected their ability to learn from humans. We hypothesized that cooperative and independent working dog breeds may behave differently in a social learning task. Dogs (N = 78 from 16 cooperative and 18 independent breeds) had to detour a transparent, V-shaped wire mesh fence. The experiment consisted of three one-minute-long trials. The control condition did not include a demonstration. In the demonstration condition, the experimenter placed a reward in the inside corner by walking around the fence. Cooperative dogs reached the target significantly faster, while independent dogs did not detour faster in trials 2 and 3 after the human demonstration. Detour latencies were not associated with the keeping conditions and training level of the subjects. As we assembled both test groups from several genetically distantly related breeds, we can exclude the explanation that higher cooperativity emerged only in particular clades of dogs; instead, functional selection for particular working tasks could enhance capacities that affect a wide range of socio-cognitive traits in dogs.

## 1. Introduction

Since their domestication [1], the most fundamental species-specific characteristic of dogs is their near-perfect and ubiquitous adaptation to the anthropogenic niche [2]. Despite great variability in the intimacy levels of dog–human interrelationships [3], one common feature of most dog populations, as well as individual dogs, is their dependency on humans and human resources [4]. The evolution of the dog involved such pressures of selection that resulted not only in a new species that can smoothly coexist with humans [5] but that also shows a strong willingness and socio-cognitive capacity to interact, learn from and cooperate with humans [6,7]. As a consequence, such species-specific behavioural and socio-cognitive phenotypes have emerged in dogs that can be considered as factors that enhance and improve dog–human coexistence and cooperation. These factors include the attachment complex between a dog and its owner [8], sociability [9], bi-directional communication, in which dogs can be both signallers and receivers (visual, [10,11]; acoustic, [12,13]), empathy, sensitivity to human ostensive communication [14], social referencing [9] and social learning [15].

Social learning is a widespread phenomenon in various taxa and species [16]; however, in almost every instance, it requires that the observer and the demonstrator are familiar with each other (i.e., they belong to the same social group) (e.g., chimpanzee (*Pan paniscus*) [17]; keas (*Nestor notabilis*) [18]; rats (*Rattus norvegicus*) [19]). More importantly, and contrary to these studies, it has been proven on many occasions that dogs can readily learn from observing human demonstrators who were previously unknown to them (e.g., [20]). This unique phenomenon provides insight to the flexibility and robustness of the dog–human relationship. We also have ample empirical evidence that dogs show social learning capacity in such contexts that are pivotal for smooth dog–human coexistence, including social anticipation [21], over-imitation [22] and the reliance on human ostensive communication during the learning phase [23].

As dogs show relatively poor performance in object manipulation tasks, in their case, some of the ‘classical’ methodologies used in social learning experiments (such as ‘artificial fruit’, e.g., [24]) are not well suited. However, spatial problem solving that does not require object manipulation is readily available for dogs; among these tasks, the detour test is notably used [25]. In a series of experiments that were based on the robust detour paradigm around a transparent V-shaped fence, Pongrácz and colleagues described many features of dogs’ social learning behaviour from a human demonstrator [15]. These results highlight that dogs exclusively rely on human-specific communication features for successful learning [23,26]; they do not restrict their attention to familiar demonstrators only [20], and they prefer a solution (detour) that has been reinforced by human demonstration, even if this is suboptimal against solutions to the problem that could be ‘discovered’ on their own (a shortcut through the fence, [27]).

It is well known that strong between-individual differences can exist both in the trial-and-error [28] and social problem-solving capacity and performance [29,30]. If social learning is one of the main mechanisms that helps dogs to adapt to their human environment, we can expect that this capacity would be fairly universal among all dogs. Supporting this theory, one study has found no difference among several dog breeds’ performance in the detour task after observing human demonstration [31]. However, we found that the social dynamics in dogs’ environments (such as their position in conspecific hierarchy [32], dog–owner interactions [33] and potentially the interaction between the environment and inherited temperament factors (personality)), could influence dogs’ social learning performance [34]. According to the results of the aforementioned studies, high-ranking dogs from multi-dog households show better social learning performance than lower-ranking individuals in case of a human demonstrator; however, lower-ranking dogs learn much better from a dog demonstrator than high-ranking dogs do [32]. Additionally, those dogs that behave assertively and show a stronger tendency to be aggressive with their owner performed better in the detour task where an unfamiliar experimenter demonstrated the task [33]. Finally, we have found that the personality traits that are associated with being more dominant in the hierarchy could also enhance dogs’ responsiveness to social stimuli in a learning context [34].

With the aim of finding traces of an inherited background underlying dogs’ socio-cognitive and behavioural phenotypes, comparative studies on various dog breeds are becoming increasingly popular (e.g., [35,36]). Unfortunately, many of the papers that deal with the comparison of dog breeds use only a handful of (e.g., [37]) or randomly choose popular dog breeds [38], which makes the finding of relevant differences difficult or means that the found differences are hard to explain within an ecologically valid framework.

From the aspect of adaptation, we would need to pinpoint such selective processes in the evolutionary past of dogs that affected canine socio-cognitive and behavioural traits that are connected to, or rely on, human behaviour. Functional breed selection for cooperative and independent working dog types [39] provides a promising clustering factor for the analysis of behaviours that involve interactions with humans. A large-scale molecular genetics study [36] found that those behaviours that show the strongest between-breed segregation belong to clusters that characterize either the cooperative (‘biddable’) or the independent working dog types.

More importantly, there are several indications that the above-mentioned clusters of working dog breeds perform differently in such tasks that involve interactions with humans. For example, cooperative dogs were found to be more successful in a visual two-way choice task based on following distal human pointing gestures [40]. In that study, the authors argued that the original task of these dogs (e.g., herding breeds, gundogs) routinely included the following of visual signals from their human handlers; thus, these dogs would be more inclined to pay attention to humans in visually communicative situations. The role of communicative signals in these findings is emphasized by later findings of [41], which showed that cooperative and independent working dogs looked at static human pictures for an equally long amount of time; thus, the differences in the earlier point-following test cannot be attributed to their different interest in humans per se. Establishing eye contact is regarded as an important initial step for interspecific communication with humans, and it was found that cooperative dogs look sooner into the eyes of a human partner [42] than independent dogs.

Interacting with humans often leads to food reward acquisition, and independent dogs seem to act with a more cost-effective strategy. In the inequity aversion paradigm, it was found that independent working breeds were more sensitive to reward omission; they stopped their interaction with the human partner sooner after the reward provision ceased [43]. In another experiment where dogs had the opportunity to ‘steal’ forbidden food, independent dogs preferred to act when the human partner could not see them take the reward, which can be considered a more successful strategy under realistic circumstances [44]. Finally, in a cognitive bias task, ref. [45] found that independent dogs showed stronger reward-maximizing tendencies and visited an ambiguous reward location more readily. The limitation of that study, however, was that it included only one independent and three cooperative dog breeds. 

While the previously mentioned experiments included contexts where dogs and humans were involved in active interactions or communicative situations, other investigations targeted the more general relationship between owners and their independent or cooperative dogs. While the complex attachment bond between a dog and its owner [8] did not show breed-group-related differences [46], in an outdoor separation test, it was found that cooperative dog breeds are more sensitive to the departure of their owner (they barked more and sooner and tried to follow the owner more eagerly, [39]). These results also indicate that cooperative dog breeds may exhibit a higher willingness to be in the vicinity of their owner and be ready for various sorts of interactive tasks with them.

### Aims of Our Current Study

According to our best knowledge, there has been no study that has found associations between the social learning performance of dogs and their breed or selection-related events in their past. A further problem is that in many studies that aimed at the associations between particular behavioural or socio-cognitive phenotypes of dogs and their inherited (breed-related) background, they used a very limited collection of breeds (often only two–three breeds), which drew broader conclusions of limited power (e.g., [37,45]). 

In our exploratory study, we used the well-known detour paradigm [25], with and without human demonstration [20], and tested dogs from the possible widest selection of breeds, which were sorted into one of the two well-established breed clusters: independent and cooperative working dogs. 

We had two hypotheses. (1) According to the first, social learning in a spatial (detour) task from a human demonstrator requires such skills as paying attention to visual cues, ostensive acoustic communication and the reward location. We hypothesized that functional breed selection (i.e., whether a dog belongs to a cooperative or to an independent breed) could affect some or all of the aforementioned skills. Therefore, we predicted that independent and cooperative dog breeds would show different performances in a social learning task. 

(2) Our alternative hypothesis was that, similar to dog–owner attachment [46], social learning from a human demonstrator would require such fundamental socio-cognitive skills that are unaffected by the (later manifested) functional breed selection. Accordingly, here, we predicted that cooperative and independent dog breeds would show no difference in their performance after observing a human demonstrator. 

Additionally, we expected that, independent of the task (trial-and-error or demonstration), cooperative dogs would look back to humans (owner, experimenter) more often when they encountered difficulty during the problem solving. Looking-back behaviour is considered to be a sign of dog–human dependency, and based on earlier findings [39,42], we assumed that cooperative working dogs would show a stronger dependency on humans than independent breeds would. 

In the case of the control condition (no demonstration), we expected no specific difference between the performances of the two functionally selected breed groups, as this task did not involve visual communication with humans. Additionally, in both groups, owners were allowed to encourage their dogs during the task. It was also found earlier that inhibitory control, thought to be important in detour tasks, has a strong inherited component across dog breeds [47]. Still, as both the cooperative and independent dog breed groups in this study consisted of functionally similar but distantly related breeds, we did not expect that the potentially strong genetic effect for inhibitory control would have an effect in the control context.

## 2. Materials and Methods

### 2.1. Subjects

We tested adult companion dogs (minimum age of 1 year, maximum age of 12 years, average ± SD = 5.03 ± 3.08 years). Dog owners were recruited through advertisements placed on social media. We specified which dog breeds we were looking for as well as the minimum age limit. We also required that the subjects had not previously participated in a detour test. Apart from these, we did not have any further requirements. Experimental groups were filled in a parallel manner, and dogs were assigned to the test and control groups randomly. 

We provide the basic demographic details of the subjects (breed, breed group, age and sex) as well as their genetic clade assignments in Table 1. When recruiting the subjects, we paid extra attention to invite representatives of both breed groups (independent and cooperative) from the widest possible range of breeds, thus avoiding the over-representation of particular breeds in any of our test groups. Therefore, we tested 18 breeds from the independent working group and 16 breeds from the cooperative working group. Additionally, we also recorded the keeping conditions of the subjects (indoor only, indoor–outdoor and outdoor only), as well as the level of training the dogs had received (none (this did not occur), training at home, course at dog school, regular dog school, assigned trainer and specific sports/work training).

### 2.2. Equipment

All tests were performed outdoors at a dog school in Budapest, Hungary. We ran all the tests between September 2022 and April 2023. We had an open, empty, approximately 10 m × 10 m fenced, grassy area at our disposal for the tests. The fenced-in area provided a calm and undisturbed environment where no other dogs or bystanders could interrupt the procedure. Our main fence equipment was identical to the one described in the article of [20]. It was a transparent, V-shaped fence made from wire mesh stretched tightly over a light steel frame. With protruding steel pegs, the fence was firmly inserted into the ground so that its lower edge was just above the soil, preventing the dogs from digging or going under the fence. The intersecting angle of the fence was set to 80 degrees. Each wing of the V-shaped fence was 3 m long, and its height was 1 m. 

We erected the fence in the middle of the area so there was at least a 3 m distance between the V-shaped fence and the perimeter of the testing area. A starting point was marked 2 m away from the corner of the V-shaped fence in the midline. We recorded the tests with two video cameras (one Canon and one Sony) that were positioned on tripods and placed to the left and right of the V-shaped fence approximately in line with the front corner. The outlay of the testing area with the V-shaped fence can be seen in Figure 1. 

### 2.3. Experimental Groups

Each subject was tested in only one experimental group, but the same owner could participate with multiple dogs in the tests. We assigned the dogs to the experimental groups by paying attention to the balanced distribution of the sex, age, keeping condition and training level of the subjects. The following experimental groups were formed:Independent dogs/control (no demonstration)—N = 17;Cooperative dogs/control (no demonstration)—N = 21;Independent dogs/detour demonstration—N = 21;Cooperative dogs/detour demonstration—N = 19.

We determined the desired sample size by using the equation for finite populations.
$$n^{\prime }=\frac{n}{1+\frac{z^{2}x\hat{p}\left(1-\hat{p}\right)}{\varepsilon^{2}N}}$$

*z* (z-score) = 1.96 for the 95% confidence level, and *ε* (margin of error) = 0.05; $\hat{p}$ (population proportion) = 0.50. We expected that the population of suitable dogs (*N*) for our test would be 100 (based on previous social media attempts with subject recruiting campaigns, this means that within a reasonable time frame, we could recruit no more than 100 dogs with their owners, in which a purebred dog truly belonged to one of the targeted breed groups). The calculated sample size was 80, and we tested a total of 92 subjects. We originally opted for slightly more dogs than the calculation suggested because we expected that some of the subjects could have possibly been excluded for various reasons. The exact details of exclusions and the actual number of excluded subjects are provided in the next section.

### 2.4. Testing Procedure

#### 2.4.1. General Procedure

Upon their arrival, dog owners (O) gave their written informed consent that they were told about the circumstances and general aims of the study. They entered the testing area accompanied by the experimenter (always the same, young woman, P.D.). The experimenter (E) explained to the O what to do, and what not to do, during the test. The dog was allowed to walk around the area on a leash; however, we did not let the dog go behind the V-shaped fence at this time. This initial ‘familiarization’ period took approximately 5 min. We asked the O whether the dog was motivated best with food or a favourite toy, and we used the reward selected by the O. 

We asked the O to position the dog on the starting point and then stand right behind it. At that point, the dog was still on a leash. The dog had to face towards the fence. The E called the dog’s attention (by calling its name and saying, for example, ‘Look’). Then, the E walked to the intersecting angle of the fence, conspicuously holding a piece of food (or the toy) in her hand, leaned over the fence and dropped the reward in the inner corner of the fence. After this, the E showed her empty hands to the dog. Then, she returned to the starting point and stood about 1 m behind the O. At that moment the O was requested to release the dog. The O’s were instructed to encourage their dogs to solve the task (i.e., to obtain the reward). Any verbal encouragements were allowed; however, we requested the O’s not to use commands such as “Forward”, “Go around” or “Go further”. Gestural commands were also disallowed. If the O broke these rules, the dog had to be excluded from the data analysis. 

The dog had 60 s to solve the task. If it performed a successful detour within the time limit and obtained the reward from behind the fence, the O had to recall the dog to the starting point, and the next trial started. If the dog did not perform a successful detour in 60 s, the trial ended., and the O had to bring the dog back to the starting point. Two consecutive trials were separated by approximately 1 min inter-trial intervals.

#### 2.4.2. Control (No Demo) Groups

In the control groups, the dogs had to perform three identical trials that were exactly the same as described in the Section 2.4.1. Each trial lasted for a maximum of 60 s, or it was shorter if the dog performed a successful detour.

#### 2.4.3. Detour Demonstration Groups

In this group, Trial 1 (baseline) was identical to the previously described (no demo) trials. However, before Trial 2 and Trial 3, the E demonstrated a detour to the dogs. In this case, the O had to position the dog back on the starting point. The dog was on a leash at this point. The E held the reward conspicuously in her hand, stepped in front of the dog and then started to walk along one wing of the V-shaped fence. While performing the demonstration, the E kept calling the dog’s attention with ostensive signals (calling the dog’s name, repeating words such as ‘Look’, ‘Here I go’, etc.). The E walked along the outside of the wing of the fence, turned in at the end and came back along the inner side of the wing towards the centre. When she arrived at the inner corner, she held up the reward for a moment and then put it down to the ground. Then, she showed her empty hands towards the dog and walked out along the other wing of the fence, still keeping the dog’s attention on herself. When the E reached the starting point and positioned herself approximately 1 m behind the O, the dog was released and encouraged to obtain to the reward. 

Trials were 60 s long. In Trial 3, the demonstration was identical to the one described in the case of Trial 2 with the exception that the E performed the detour from the opposite direction (i.e., if Trial 2 had a left-to-right detour direction, in Trial 3, the E walked right-to-left). 

The direction of the detour demonstration in Trial 2 was based on the direction of the dog’s successful detour in Trial 1 because the E always started the demonstration on the opposite side of the fence to that which the dog used in Trial 1. In the case of an unsuccessful Trial 1 (i.e., the dog did not have a successful detour in Trial 1), the E randomly chose the side to use in Trial 2 for the demonstration. 

In the detour demonstration groups, the O had to adhere to the exact same instructions as they did in the control groups. 

### 2.5. Exclusions

We excluded subjects that were not motivated to perform any trials or lost interest for further performance during the test. A dog was considered to have lost interest if it did not approach the V-shaped fence upon its release from the starting point or only approached it once. We had to exclude N = 12 dogs altogether for this reason. Their distribution in the four testing groups was as follows: cooperative/control N = 1; cooperative/demo N = 1; independent/control N = 6 and independent/demo N = 4. These non-motivated dogs had the following training backgrounds: training at home N = 2; course at dog school N = 4; regular attendee to dog school N = 2; assigned trainer N = 2 and specific work/sports training N = 2.

We had to exclude an additional dog because it was too afraid of the test situation and the experimenter; independent/control N = 1. Another dog was excluded because it got loose when the experimenter performed the demonstration; cooperative/demo N = 1. The results of the excluded dogs did not appear in the statistical analysis. 

### 2.6. Behavioural Coding

Each test was video recorded. We used Solomon Coder (beta 19.08.02, copyright by András Péter) for the extraction of data from the video sequences. Table 2 shows the behavioural variables we used for the analysis. Most of these appeared in [20], but we also developed new variables because we wanted to provide a more detailed description of the dogs’ behaviour during the tests. The reason for this was that we expected the difference between the two breed groups to be potentially a minor one; thus, we prepared to detect it with more meticulous behavioural coding. For the inter-coder reliability analysis, 10 percent of the videos were re-coded by a second experimenter who was unaware of the breed group assignment of the subjects and the experimental hypotheses.

### 2.7. Statistical Analyses

Success rates (whether the dog reached the reward or not in the given trial, 1 vs. 0) were analysed with GEE (general estimating equations with binary logistics). In the analysis, dog ID served as a random factor, experimental group (1–4) served as a fixed factor and trial (1–3) served as a repeated factor. 

Latencies were analysed with Cox’s regression models. Latencies of turning at the rear end of one of the wings, as well as latencies of obtaining the reward, were analysed separately. Across the groups, we compared all Trial 1 (baseline) latencies to see whether any of the groups had inherently faster or slower problem-solver dogs (Trial 1 was without a detour demonstration in each case). To test whether environmental factors would also affect the dogs’ performance, we added the sex of the dog, keeping condition and training level as fixed factors, along with the breed group, to this model. We ran similar models with all Trial 2 and Trial 3 turn-latencies and reward-latencies to see the potential effect of the keeping conditions and training level of the dogs. 

To test for the effect of social learning, in separate Cox regression models, we analysed whether cooperative and independent working dogs improved the speed of detouring (measured with the turn-latencies and reward-latencies) along the consecutive trials in the treatment groups (with a demonstration). Latencies of consecutive trials were separately compared within the control (no demonstration) groups as well to see whether the mere repetition of trials (thus, trial-and-error learning) would cause an improvement in the detour efficiency. 

We analysed the concordance of the dogs’ detours in relation to the direction of their successful attempt in Trial 1 with Wilcoxon’s signed-rank tests (in both the control and detour demo groups). We only included the dogs in this analysis who successfully performed the detour in Trial 1. Similarly, Wilcoxon’s signed-rank tests were applied to analyse whether the direction of the detour by the demonstrator had any influence on the dog’s subsequent performances (only in the detour demo groups). In these analyses, we used the number of subsequent trials in which concordance occurred either with the direction of the dog’s first successful detour or with the direction of the demonstrator. The hypothetical median of concordance was set to 1 (min = 0; max = 2) in the case of both the concordance with previous own detours and with the demonstrator’s side choice. 

Task focus duration, the frequencies of side alternations, looking back at the humans and the owner’s encouragement were analysed with mixed general linear models (with trials (1–3) as a repeated factor and experimental treatment and breed group as fixed factors). 

To check the reliability of the coding method, an independent observer (who was unaware of the test hypotheses) coded video footage from 10 randomly chosen dogs. Latency and frequency data were analysed by Spearman’s rho correlation. Based on the analysis, our coding procedure was reliable (reward latency: R_(23)_ = 0.995; *p* < 0.001; *p* < 0.001; ‘Leave’ duration: R_(30)_ = 0.994; *p* < 0.001; ‘Encouragement’ frequency: R_(30)_ = 0.993; *p* < 0.001; ‘Lookback’ frequency: R_(30)_ = 1.00; *p* < 0.001; ‘Side-alternation’ frequency: R_(30)_ = 1.00; *p* < 0.001). All statistical analyses were performed in SPSS.22.

## 3. Results

In the case of the frequency of successful detours (when the dog managed to detour the fence and obtain the reward), we found a significant effect of the repeated factor (GEE with binary logistics; trials: (χ^2^_(2)_ = 10.410; *p* = 0.005)), however, the effect of the breed group was not significant (χ^2^_(3)_ = 2.564; *p* = 0.464). The success rate of the dogs became higher along the consecutive trials regardless of the breed group or the experimental treatment (demonstration or control). 

The latencies of turning at the rear end of the wings of the fence and reaching the reward provided similar results in the between- and within-group Cox regression analyses. We did not find any significant effect of the sex of the dog, the keeping conditions and the training level on the two types of latency data, and the latencies of Trial 1 across the experimental groups did not differ significantly (Table 3).

When we compared the detour latencies within the breed groups, we found a significant effect of repeated trials in the case of the cooperative working dog group, but only in the case of the human demonstration treatment. No significant improvement in detour latencies was found in Trials 2 and 3 in the case of the human demonstration for the independent working dogs (Figure 2). Neither of the control groups improved their detour latencies across the trials (Figure 3). These results were very similar in the case of both types of latencies (Table 4, turning at the rear end of the fence and reaching the reward).

Regarding the relative task focus durations (i.e., the dog left the vicinity of the fence during the trial), we did not find any significant effect in the case of the breed groups (F(1, 59) = 0.002; *p* = 0.968), test condition (F(1, 59) = 1.168; *p* = 0.284) and repetition of the trials (F(2, 118) = 0.887; *p* = 0.415). 

The frequency of the owners’ encouraging utterances during the tests showed a significant effect of the repeated trials (F(2, 150) = 3.521; *p* = 0.032; they encouraged the dogs less frequently in Trial 3 than in Trials 1 and 2). Breed group (F(1, 75) = 3.203; *p* = 0.079) and test condition (F(1, 75) = 0.103; *p* = 0750) did not have significant effect on the frequency of encouraging the dog (Figure 4).

The frequency of looking at the humans during the test showed a significant effect of the repeated trials (F(2, 150) = 3.761; *p* = 0.025; the dogs looked at the humans more frequently in Trial 1 than in Trials 2 and 3); however, neither test condition (F(1, 75) = 0.286; *p* = 0.594) nor the breed group (F(1, 75) = 1.468; *p* = 0.229) had a significant effect on it (Figure 5). 

The frequency of side alternations at the corner of the fence did not show any association with the repeated trials (F(2, 150) = 2.658; *p* = 0.073), test condition (F(1, 75) = 0.803; *p* = 0.373) and breed groups (F(1, 75) = 0.796; *p* = 0.375). 

Finally, with the one-sample Wilcoxon signed rank tests, we checked whether the dogs followed either their own direction from Trial 1 in the subsequent trials or the demonstrator’s (inward) side choice in the two demonstration groups. After performing Bonferroni’s correction for multiple comparisons (adjusted *p* = 0.008), concordance with the dogs’ own side choice and the demonstrator’s direction did not differ from the hypothetical value (=1) in any of the groups. Own direction: cooperative dogs/control (T = 1.508; N = 13; *p* = 0.132); cooperative dogs/demonstration (T = −1.633; N = 9; *p* = 0.102); independent dogs/control (T = 0.000; N = 9; *p* = 1.000); independent dog/demonstration (T = 0.955; N = 15; *p* = 0.366). Demonstrator’s direction: cooperative dogs/demonstration (T = 0.000; N = 11; *p* = 1.000); independent dogs/demonstration (T = −2.121; N = 16; *p* = 0.034). In other words, the dogs did not follow the demonstrated side while attempting to detour the fence, and they also did not stick to their own choice in Trial 1. 

## 4. Discussion

The main goal of this research was to see whether observing a human demonstrator’s action would be equally helpful for all sorts of working dog breeds or whether their specific relationship with humans, due to function-based selective pressure in their past, would differently affect their capacity of utilizing a human-provided behavioural template. In the case of the latter, we could better pinpoint those factors that enable dogs to learn from humans in a spontaneous (i.e., non-training) situation, which in turn would shed more light on the complex system of effective dog–human coexistence and cooperation. In our experiment, using the robust detour paradigm around a V-shaped transparent fence, we showed that dogs belonging to cooperative working breeds benefited from observing a human demonstrator who showed them how to effectively master the detours. However, while the cooperative working dogs performed the detours faster after the demonstration compared to their own first (baseline) trial latencies, the dog breeds that were originally selected for independent working tasks showed no improvement in their detouring speed after observing the same demonstration. More importantly, independent working dogs could solve the task with the same frequency of success as the cooperative dogs; thus, the difference between their post-demonstration behaviours was clearly attributable to their relatively elevated speed due to learning from the human demonstrator. We found no difference between the problem-solving efficacies of the breed groups when they had to rely on trial-and-error learning in the control (no-demonstration) conditions, as none of the groups showed improvement in their speed in the consecutive control trials. Neither the keeping conditions nor the training level of the dogs had an effect on their problem-solving performance.

Recently, using dog breeds as the main explanatory variable in behavioural sciences has become a popular and often-pursued endeavour. Scientists keenly rely on the genetically more-or-less closed sub-populations of dogs (called ‘breeds’) when they target the behavioural correlates of domestication in relationship with divergence from their hypothetical wolf-like ancestor [51] or when they try to tackle the fundamental changes in behaviour due to the hypothetical ‘domestication syndrome’ (e.g., [52]). Breed-related behavioural differences have also become an important target for researchers since the impact of particular problematic behaviours (among others, inter- and intraspecific aggression [53] and separation-related problems [39]) was recognized to have a major impact on the welfare of both dogs and humans. Finally, the potential difference between dog breeds’ behaviour and cognition can also provide important insight into particular socio-cognitive phenotypes that have relevance in comparative cognition research, including various neuro-behavioural conditions in humans (e.g., ADHD-like behaviour: [54]; autism spectrum disorder, [55]).

In our opinion, one of the most important, still often-missed, criteria of posing a question about breed-related behavioural differences in dogs is the ecological/biological validity of the researchers’ goal in the chosen framework of subjects. Perhaps the most-often-followed strategy is to compare the behavioural traits of a smaller or larger set of ‘most popular’ dog breeds [38,56,57,58]. Although it often provides intriguing results, this approach usually lacks a priori hypotheses about the factors that could explain the eventually found differences. On the other hand, when researchers pre-set their hypothesis about the targeted breed-related behaviours, it still happens that they compare very few (often only one per category) dog breeds while still claiming that the found differences could be attributed to a broader factor behind those breeds (e.g., [37,45]). This is why in our investigation we opted for functional breed selection as our grouping variable, which is probably one of the most promising and validated explanatory factors in canine cognitive ethology (e.g., [36,39,40]), and, more importantly, we put strong emphasis on testing a high number of different dog breeds in both groups without the overrepresentation of any popular breeds within them.

Our most important finding fell in line with our prediction, namely that dog breeds that were selected for cooperative work with humans (e.g., herding dogs, gundogs, retrievers) learned to detour around the fence and reach the reward faster when they had the opportunity to observe a human demonstrator before the trial. While independent working breeds (e.g., terriers, hounds, sled dogs) detoured the fence with similar latencies compared to the cooperative dogs in the first trial, they did not improve their detouring speed after observing the demonstrator. According to our best knowledge, this is the first time that a breed-related difference has been found in a social learning task where dogs were provided with a human demonstration (see earlier negative results, [31]). 

By choosing functional breed selection as a grouping variable, we targeted such capacities in our subjects that could provide an overarching effect across many dog breeds that may only show distant genetic relatedness at the same time. Based on molecular genetic data, our cooperative and independent dog breeds were not necessarily closely related [48] (see also Table 1). For example, sight hounds, terriers and sled dogs are each independently working breeds, but they fall far from each other in the cladogram that was established by [48]. Similarly, many breeds within the cooperative working dog group are positioned at very different branches of the cladogram (e.g., Collie-types, Hungarian herding dogs and retrievers, [48]). Unlike earlier studies that investigated behavioural phenotypes based on genetically related groups of dogs (e.g., boldness and trainability: [59]; low attachment and attention-seeking behaviour in ‘ancient’ and spitz-type dog breeds, [60,61]), our results could highlight shared, ecologically valid factors among dog breeds that underwent such selective processes that resulted in functionally similar socio-cognitive capacities.

The recent article by [36] found a well-established genetic basis for those behavioural traits that characterize ‘biddability’ (i.e., responsiveness to human direction and commands)—an important feature shared by cooperative dog breeds. Earlier research also found that cooperative dog breeds performed better in tasks where human visual communicative cues could lead the dog to a hidden reward in a two-way choice test [40]. In light of these facts, in our detour test where the human demonstrator showed the solution to the canine subjects, what could the decisive components that made cooperative dogs the more effective learners be? The demonstrator’s behaviour provided multiple clues: a scent trail, the sight of the human moving around the fence, the target was also visibly being carried around, and, finally, the human demonstrator grabbed the dogs’ attention with ostensive verbal cues. All of these elements were equally present in the case of both groups during the demonstration. In an earlier study, ref. [23] ruled out the decisive effect of all but one of these factors. They found that the most important feature that made the human demonstration truly effective was the demonstrator’s ostensive communication during the detour. Without this, an otherwise complete demonstration remained ineffective for the dogs. Currently, we do not have evidence about the potentially different attentiveness behaviours towards ostensive verbal signalling in cooperative and independent dog breeds; thus, it remains unknown whether the cooperative dogs were more keen to observe and learn from the human demonstrator because of their more intense interest in her ostensive signals or whether they observed the visual component of the demonstration with a stronger interest. As the dog breeds in our experiment were mostly selected for visually cooperative or independent tasks, and the earlier results were also connected to visual signalling (i.e., point following, [40]), one can intuitively assume that the independent dogs had a harder time (or less interest in) following the demonstrator’s walking pattern. Summarizing the found differences between the cooperative and independent dog breeds’ social learning performance, we consider the functional selection for cooperative work with humans to be a decisive factor behind the results. Cooperative breeds score high on such genetically determined features as ‘biddability’ and ‘easy to train’ [36], meaning both features are likely to enhance and ease following and learning from a human demonstrator. 

One could argue that the effect of human demonstration (versus the lack of improvement in the control groups) could be caused by the longer handling time of the reward during the detour demonstration compared to the shorter route the experimenter walked with the reward in hand in the control conditions. However, this does not explain why the independent dogs did not learn while they were observing the demonstration. Potentially, the difference between the two groups’ responses to the human demonstration could be explained with such environmental factors as the owners’ different dog keeping and training habits depending on what breed of dog they had. However, it is important to see that our results were not confounded by such often-encountered and potentially influential factors such as indoor vs. outdoor keeping conditions [62], level of training [63,64] and over-representation of particular dog breeds in the sample [45]. 

Neither of the two groups’ performances improved in the control condition—this result was in agreement with several earlier papers (e.g., [20,23,32]). Detouring around a transparent V-shaped fence is a difficult task for dogs, and they are seemingly unable to improve on their own. Now, we can assume that functional breed selection did not affect this sort of spatial problem-solving capacity in any of the groups. Although inhibitory control (an important feature in any detour task as the subject should inhibit its inclination to approach the target through the shortest way, [25]) was shown as having a strong genetic predisposition across dog breeds [47], by testing nearly twenty different breeds in each group, we reliably showed that the independent and cooperative dogs were of an equally low efficiency in the trial-and-error version of our test. This result indirectly emphasizes the role of social learning from the human demonstrator as the main factor leading to the difference we found between the two groups in the demonstration condition. The cooperative dogs were not inherently more apt to solve the detour tasks by themselves than the independent breeds were, and we also showed that it was not their higher level of training that resulted in their better performance in the demonstration condition. These results were in parallel with the difference that was found between the higher- and lower-ranked dogs by [32]. In that study, dominant and subordinate dogs performed with different efficacies in the demonstration conditions; however, both were equally unsuccessful in the (no-demo) control context. 

Although we predicted that cooperative dogs would more frequently look at the humans during the detour task, we actually found that the two breed groups did not differ from each other in this aspect. Instead, the dogs in both groups looked significantly less frequently at the human in their subsequent trials compared to the first one. Looking at the nearby human is strongly associated in dogs when encountering a difficult or unsolvable problem [64,65], and this was also confirmed several times in detour tests, where the less successful the dogs were, the more frequently they looked at the humans (e.g., [20,27]). Our current results still fell in line with these results, as the dogs from both breed groups solved the detour problem somewhat more successfully in the last two trials than in the first one, whilst their looking-back frequencies dropped accordingly. There is an interesting question however: why did the independent dogs not look back at the humans more frequently, whereas in the demonstration group their detour latencies did not improve compared to the results of the cooperative dogs? There are at least two explanations for this: (1) Parallel with our prediction, the independent dogs looked at the humans less, even if they were encountering difficulties. Alternatively, (2) a slower detouring speed did not indicate that the independent dogs would ‘have had a problem’ with the task. As the frequency of successful detours was similar in the two breed groups across the conditions, in the long run, based on the number of successful detours, the independent dogs were similarly as successful as the cooperative dogs in solving this task; it was the speed of success that was the main differentiating factor.

It is worth mentioning that particular dog breeds comprise already distinct ‘working’ and ‘show’ lines, which are under different pressures of artificial selection. In the future, it would be interesting to see whether dogs from these lines would perform differently in the experiments that we described here. In addition, although we found no significant effect of the dogs’ training levels, one could hypothesize that particular forms or types of training could still influence the trial-and-error or social learning capacity of dogs. Time and further experiments can only tell whether such factors would be relevant in the case of dogs’ social learning from humans.

## 5. Conclusions

Social learning from humans is a fundamental element of dogs’ adaptive skillset to the anthropogenic niche. With our new experiment, we gathered, for the first time, evidence that dog breeds can be associated with differences in social learning performance from a human demonstrator. However, for this, it was necessary to find an ecologically valid approach, which we managed through a comparison of functionally different breeds, namely independent and cooperative working dogs. Functional breed selection could have an overarching effect across dog breeds that are genetically far from each other, those that have developed in various locations and those developed for various working tasks that either required them to be in regular visual feedback with their handler or where they had to solve their task on their own. Because of this approach, our results were not a consequence of the close genetic relationship between similarly behaving dog breeds; they refer to targeted socio-cognitive traits during their functional selection. 

Cooperative dog breeds, regardless of their ancestry and training levels, utilize human demonstration more effectively. We speculate that they might be more sensitive to ostensive communication than independent dogs; however, more parsimoniously, we should assume that the visual component of the demonstration is more relevant for them as well. 

Our results indicate that relatively recent functional breed selection can influence such fundamental socio-cognitive capacities in dogs that are necessary for effective social learning from humans and, in a broader sense, most probably contribute to the most fundamental socio-cognitive skills that dogs need for effective dog–human interactions. 

## Figures and Tables

**Figure 1 animals-13-02001-f001:**
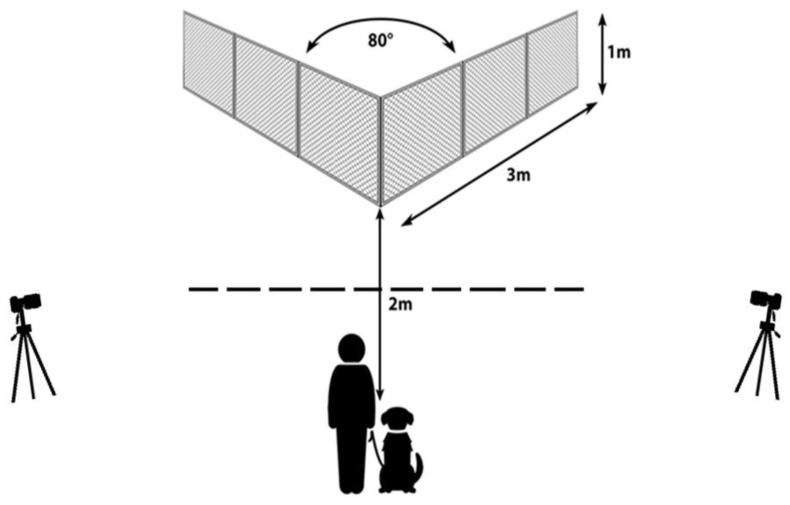
The V-shaped fence with the owner and the dog at the starting point. The reward was always placed to the inside corner of the fence at 20 cm away from the corner. We used two video cameras on tripods to record the tests (approximate position of the cameras is indicated). The dashed line shows the boundary between the areas that we considered either as the ‘vicinity of the fence’ or away from it (this was important for coding ‘task focus duration’).

**Figure 2 animals-13-02001-f002:**
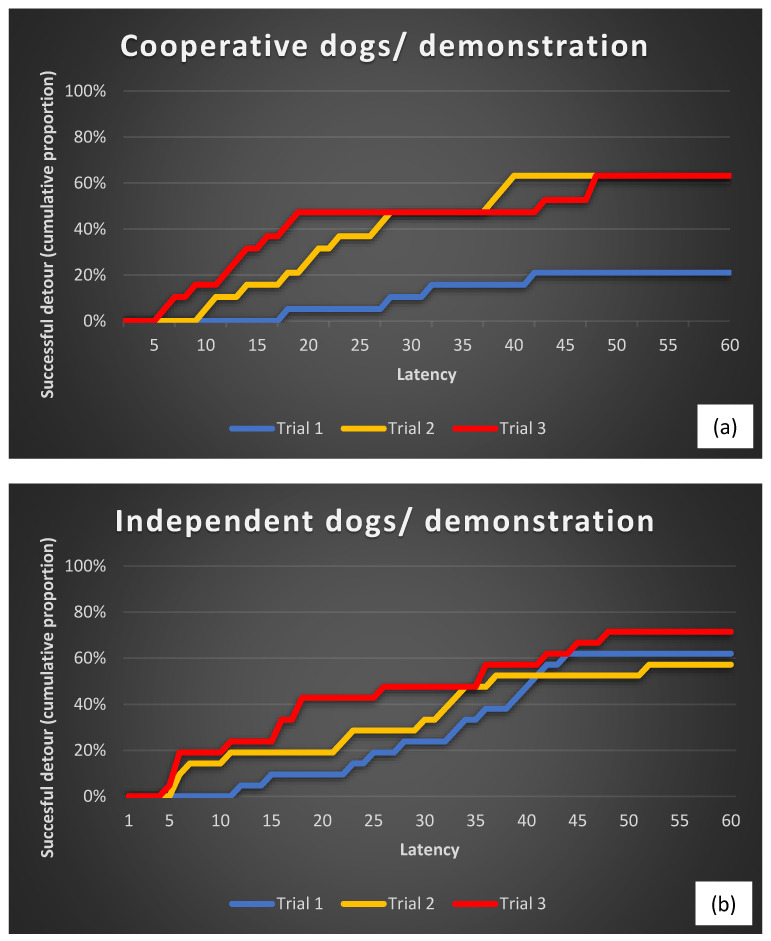
Cumulative proportions of cooperative (**a**) and independent (**b**) working dogs that performed a successful detour (reaching the reward) in the human demonstration condition.

**Figure 3 animals-13-02001-f003:**
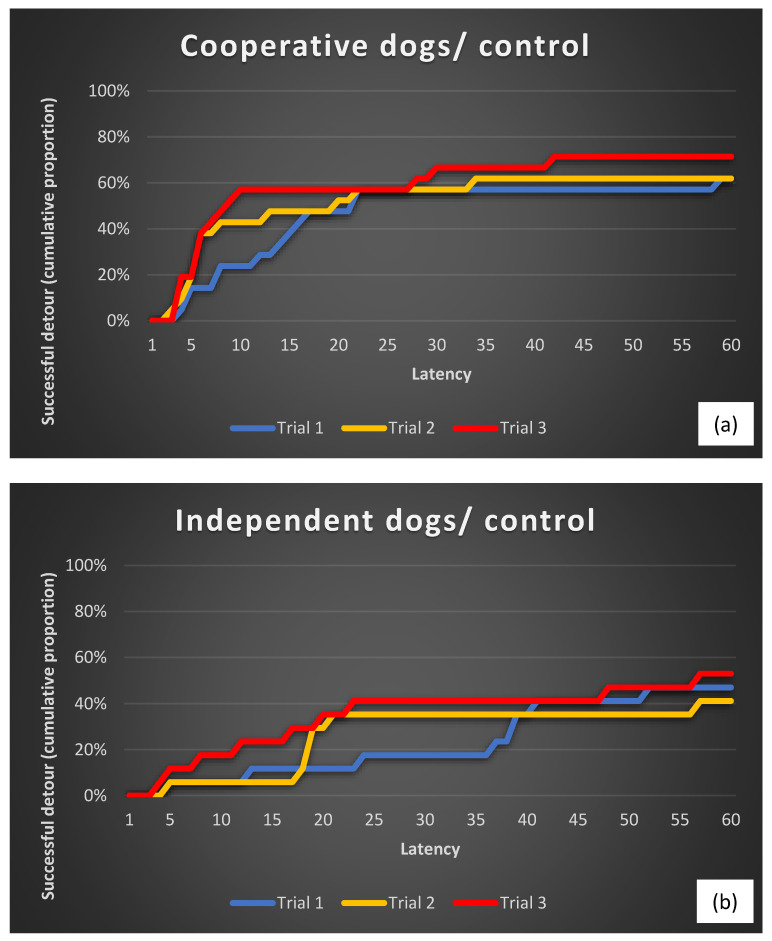
Cumulative proportions of cooperative (**a**) and independent (**b**) working dogs that performed a successful detour (reaching the reward) in the control (no demonstration) condition.

**Figure 4 animals-13-02001-f004:**
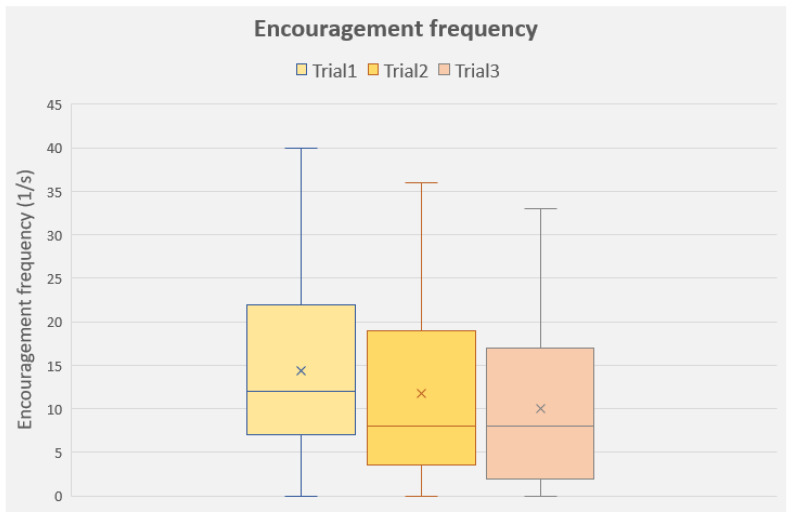
Encouragement frequencies of the dog owners during the three trials (all groups together). Median, lower and upper quartiles and minimum and maximum values.

**Figure 5 animals-13-02001-f005:**
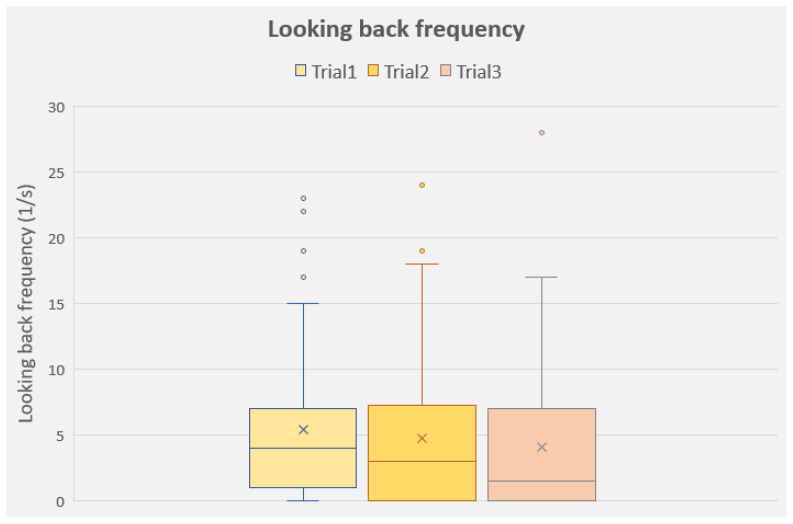
Frequency of looking back at the humans (owner and experimenter) during the three trials (all groups together). Median, lower and upper quartiles and minimum and maximum values.

**Table 1 animals-13-02001-t001:** List of the participating dogs whose data were included to the statistical analyses. Age is given in years. We indicated test group assignments as well. Coop-d = cooperative dogs with human demonstration; Coop-n = cooperative dogs without demonstration; Ind-d = independent dogs with demonstration; Ind-n = independent dogs without demonstration. Genetic clade assignments of the dog breeds were mostly based on [48], with exception of * = [49]; ** = [50], *** = https://embarkvet.com/resources/dog-breeds/spanish-galgo/ (accessed on 1 May 2023); **** = https://embarkvet.com/resources/dog-breeds/yakutian-laika/ (accessed on 1 May 2023); n/a = no available data for genetic clade assignment.

Dog’s Name	Breed	Sex	Age	Test Group	Clade
Bukfenc	German Shepherd Dog	female	5	coop-d	M
Ferenc	Mudi	male	5	coop-d	G **
Grabovszky	Border Collie	male	2	coop-d	T
Grimbusz	Pumi	male	4	coop-d	G
Jar Jar	Hungarian Vizsla	female	10.5	coop-d	R
Kifli	Hungarian Vizsla	male	8	coop-d	R
Márkó	Labrador Retriever	male	2	coop-d	Q
Matcha	Bouvier des Flandres	female	1	coop-d	S
Merlin	Border Collie	male	2	coop-d	T
Millie	Collie	female	11	coop-d	T
Mimo	Australian Shepherd	female	3	coop-d	T
Nala	Border Collie	female	2	coop-d	T
Piszke	Pumi	male	4	coop-d	G
Szeder	Border Collie	female		coop-d	T
Szille	Golden Retriever	female	4	coop-d	Q
Szotyi	Mudi	male	1	coop-d	G **
Whisky	Golden Retriever	male	2.5	coop-d	Q
Winnie	Australian Shepherd	male	4	coop-d	T
Zuzu	Australian Shepherd	female	3.5	coop-d	T
Aeon	Shetland Sheepdog	male	5	coop-n	T
Annabell	Mudi	female	5	coop-n	G **
Arwen	Border Collie	female	2.5	coop-n	T
Blue	Weimaraner	male	5.5	coop-n	R
Chuck	Briard	male	4.5	coop-n	S
Enid	Border Collie	female	6.5	coop-n	T
Fox	Cardigan Welsh Corgi	male	5.5	coop-n	T
Funky	Labrador Retriever	female	10	coop-n	Q
Golyó	Border Collie	male	3	coop-n	T
Lego	Golden Retriever	male	10	coop-n	Q
Menyus	Mudi	male	11	coop-n	G **
Murci	Mudi	female	2	coop-n	G **
Nala	Border Collie	female	5	coop-n	T
Negro	Labrador Retriever	male	4	coop-n	Q
Nugát	Labrador Retriever	female	5	coop-n	Q
Phoebe	Lagotto Romagnolo	female	1.5	coop-n	R *
Rege	Shetland Sheepdog	female	8	coop-n	T
River	Tervueren	male		coop-n	S
Sam	Briard	female	3	coop-n	S
Sydney	Australian Shepherd	male	2	coop-n	T
Szendi	Mudi	female	5.5	coop-n	G **
Aibell	Irish Terrier	female		ind-d	L
Alfréd	Dachshund	male	5	ind-d	O
Bodza	Transylvanian Hound	female	8	ind-d	n/a
Boreas	Hovawart	male	4	ind-d	n/a
Caci	Transylvanian Hound	male	4	ind-d	n/a
Dio	Pitbull Terrier	female	12	ind-d	n/a
Dongó	Transylvanian Hound	male	5	ind-d	n/a
Frida	Basset Hound	female	9	ind-d	O
Hattyú	Transylvanian Hound	female	3	ind-d	n/a
Hota Hota	Galgo	male	6	ind-d	T ***
Indiana	Hovawart	male	4.5	ind-d	n/a
Lucas	Yakutian Laika	male	1	ind-d	A ****
Málna	Dachshund	female	7	ind-d	O
Maya	Cairn Terrier	female	7	ind-d	L
Müzli	Fox Terrier	female	5	ind-d	L
Norton	Dachshund	male	4	ind-d	O
Patrik	Dachshund	male	8	ind-d	O
Pimasz	Fox Terrier	male	9	ind-d	L
Rowan	Yakutian Laika	male	2	ind-d	A****
Tisza	Hovawart	female	12	ind-d	n/a
Zete	Transylvanian Hound	male	1	ind-d	n/a
Avatár	Galgo	female	7.5	ind-n	T ***
Csoki	Dachshund	male	11	ind-n	O
Dió	Jack Russell Terrier	male	3.5	ind-n	L
Dorka	Komondor	female	7.5	ind-n	N
Figaro	Fox Terrier	male	3	ind-n	L
Ikon	Transylvanian Hound	male	6	ind-n	n/a
Júlia	Hungarian Greyhound	female	3	ind-n	T
Mamba	Pitbull Terrier	female	9	ind-n	n/a
Múú	Cane Corso	male	6	ind-n	W
Norwin	Irish Terrier	male		ind-n	L
Olga	Basset Hound	female	1	ind-n	O
Rhysand	Yakutian Laika	male	1	ind-n	A ****
Rudi	Hovawart	male	1.5	ind-n	n/a
Szofi	West Highland White Terrier	female	1	ind-n	L
Szvetlana	Borzoi	female	1	ind-n	T
Temida	Polish Greyhound	female	5	ind-n	T
Zack	Siberian Husky	male	5.5	ind-n	A

**Table 2 animals-13-02001-t002:** The list of behavioural variables that we extracted from the video footage of the tests. We indicated whether the particular behavioural variable was originally used in [20] or it was newly developed for this study.

Behavioural Variable	Unit	Description
Success [20]	Occurrence (0–3)	The dog reached the reward after performing a successful detour; then, it touched/consumed the reward.
Reward (detour) latency [20]	(s)	The time elapsed between the moment of releasing the dog by the owner at the starting point and the dog’s arrival at the reward (i.e., after a successful detour). In the case of an unsuccessful trial, 60 s was assigned.
Turn (detour) latency (NEW)	(s)	The time elapsed between the moment of releasing the dog by the owner at the starting point and the dog’s arrival at the rear end of one of the wings of the fence. ‘Arrival’ happened when the dog turned in at the rear end of the fence.
Detour direction (inward) [20]	Left or right or 0	The side of the fence where the dog went in its preceding successful detour attempt. In unsuccessful trials, 0 was assigned.
Concordance (own direction) [20]	1 or 0 per trial	Whether the dog performed the detour (inward) in Trials 2–3 on the same side that it used in Trial 1.
Concordance (demo direction) [20]	1 or 0 per trial	Whether the dog performed the detour (inward) in Trials 2–3 on the same side that the demonstrator used.
Task focus duration (NEW)	total duration per trial/detour latency	Task focus describes the dogs’ tendency to leave the close vicinity of the fence during its attempts to detour around it. The dog stepped over the boundary line drawn 1 m from the corner of the fence while moving away from the fence and back towards the owner. The value was calculated as the total time when the dog was away from the fence divided by the reward detour latency.
Looking back [20]	1/s	During attempts to detour, the dog turned towards the owner/experimenter (by turning its head only or with full body orientation) and looked at them. The number of looking back events was then divided by the reward detour latency.
Side alternation (at corner) [20]	1/s	The number of swapping the side events at the corner of the fence during the dog’s attempts to detour divided by the reward detour latency.
Encouragement (by owner) [20]	1/s	The number of distinct verbal utterances (at least 1 s between two) given by the owner during the dog’s attempts to detour divided by the reward detour latency.

**Table 3 animals-13-02001-t003:** Results of the Cox’ regression analysis in the case of Trial 1, Trial 2 and Trial 3 for the latencies of turning at the end of one of the wings of the fence and latencies of reaching the target.

Dependent Variable	Trial	Variable (Fixed Factors)	Chi-Square	Df	*p*
Turn (detour) latency	1	Groups	5.932	3	0.115
		Keeping	0.914	2	0.633
		Training	0.235	4	0.994
		Sex	0.100	1	0.752
	2	Groups	3.163	3	0.367
		Keeping	4.481	2	0.106
		Training	2.091	4	0.719
		Sex	0.007	1	0.931
	3	Groups	3.154	3	0.369
		Keeping	2.973	2	0.226
		Training	1.779	4	0.776
		Sex	0.002	1	0.967
Reward latency	1	Groups	5.724	3	0.126
		Keeping	1.087	2	0.581
		Training	1.697	4	0.791
		Sex	0.203	1	0.652
	2	Groups	2.454	3	0.484
		Keeping	5.272	2	0.072
		Training	3.069	4	0.546
		Sex	0.050	1	0.823
	3	Groups	3.083	3	0.379
		Keeping	3.026	2	0.220
		Training	2.016	4	0.733
		Sex	0.002	1	0.963

**Table 4 animals-13-02001-t004:** Results of the Cox regression analysis in the case of the two breed groups of dogs and the two test conditions (Control = without human demonstration; Demonstration = the experimenter showed the detour to the dogs before Trial 2 and Trial 3). Significant effects are highlighted with bold letters.

Dependent Variable	Breed Group	Test Condition	Chi-Square	df	*p*
Turn (detour) latency	**Cooperative**	Control	0.715	2	0.699
		**Demonstration**	**9.069**	**2**	**0.011**
	Independent	Control	0.606	2	0.738
		Demonstration	1.395	2	0.498
Reward (detour) latency	**Cooperative**	Control	0.949	2	0.622
		**Demonstration**	**8.814**	**2**	**0.012**
	Independent	Control	0.598	2	0.742
		Demonstration	1.490	2	0.475

## Data Availability

Raw data are supplemented in the electronic Appendix A.

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
