# Peer review of "Would You Detour with Me? Association between Functional Breed Selection and Social Learning in Dogs Sheds Light on Elements of Dog–Human Cooperation"

_animals, 2023, doi:10.3390/ani13122001_

Round 1
Reviewer 1 Report
The manuscript “Would You Detour with Me? Association between Functional Breed Selection and Social Learning in Dogs Sheds Light on Ele-Ments of Dog-Human Cooperation” is interesting, clear and well written. The experimental design is appropriate and allows studying the effects of breed groups on social learning in dogs.
I have only two major comments
First, as far as I understand, during the demonstration the E touched the reward twice but during the control he touched it once. This could explain the results as a consequence of a higher motivation related to a stimulus enhancement process. You should address this topic in the discussion.
Second, the results of figure 2 are intriguing. It seems that the independent group performed better than the cooperative one from the beginning. Therefore, it is possible that they did not benefit from the demonstration due to a ceiling effect. Did you analyze a possible effect of the interaction between breed and trial on the performance of the demonstration groups? Remarkably, this trend is not observed in the control group.
Some minor comments:
L 84-88 considering their relevance, it would be enriching if you briefly describe those findings.
L 148 as you stated the two possible hypotheses; you might clarify the exploratory nature of the study
L 178 add maximum
I think that it is important to analyze possible effects of sex, age and neutered status.
Please, include in Figure 1 the position of the cameras.
It is necessary to mention the ethics considerations.
Did a second experimenter hold the leash?
How long was the familiarization time?
How many owners did not follow the instructions?
Please, clarify if dogs received any reward when they went back to the starting point during failed trials. It is important in order to evaluate the loss of motivation throughout trials.
In the demonstration groups it would be clearer if you describe the first trial as a baseline.
Which was the inter-trial interval?
The colors of the figures are difficult to see, I suggest using darker ones.
It would be more informative if you show the performance of each group in Figures 4 and 5.
Based on your conclusion that independent breeds detoured the fence with similar latencies compared to the cooperative dogs in the first trial but they did not improve their detour speed after observing the demonstrator, I would expect differences between them in the third trial.
L 549 May be a useful measure would be the frequency of approaching the vertex first.
L 598, please, replace the word “suspect”.
Author Response
RESPONSES TO Reviewer 1
The manuscript “Would You Detour with Me? Association between Functional Breed Selection and Social Learning in Dogs Sheds Light on Ele-Ments of Dog-Human Cooperation” is interesting, clear and well written. The experimental design is appropriate and allows studying the effects of breed groups on social learning in dogs.
RESPONSE: Thank you for the supportive report and for the thoughtful recommendations.
I have only two major comments
First, as far as I understand, during the demonstration the E touched the reward twice but during the control he touched it once. This could explain the results as a consequence of a higher motivation related to a stimulus enhancement process. You should address this topic in the discussion.
RESPONSE: Thank you for this interesting comment. In reality, the experimenter handled the reward similarly both in the control and in the demo conditions. This means that she carried the reward to the fence, holding it visibly in her hand, then showed it up, then put (dropped) it on the ground, showed her (empty) hands again, then returned to the dog (and owner). There is a difference though in the duration of the two procedures: the experimenter carried the reward for a shorter time in case of the control condition than in the demonstration groups. However, the independent dogs still did not improve when they could observe the demonstration. Additionally, in our earlier experiment (Pongrácz et al., 2004), we found that simply carrying the reward around the fence, without maintaining the dogs’ attention with ostensive signals, makes the demonstration ineffective. We have added a few sentences about this question to the Discussion, which reads like this: “One could argue that the effect of human demonstration (versus the lack of improvement in the control groups) could be caused by the longer handling time of the reward during the detour demonstration compared to the shorter route the experimenter walked with reward in hand in the control conditions. However, this does not explain why not the independent dogs learned while they were observing the demonstration.”
Second, the results of figure 2 are intriguing. It seems that the independent group performed better than the cooperative one from the beginning. Therefore, it is possible that they did not benefit from the demonstration due to a ceiling effect. Did you analyze a possible effect of the interaction between breed and trial on the performance of the demonstration groups? Remarkably, this trend is not observed in the control group.
RESPONSE: Thank you for this observation. We agree that the baseline latency of the cooperative dogs in the demo group looks higher than the baseline for the independent demo group. However, statistically we did not find an effect of group on Trial 1 latencies. Similarly, performance did not show breed x trial interaction effect. The ceiling effect in the independent dogs’ group with demonstration is an interesting question. We checked the mean Trial 1 latency in the independent dog group with demonstration (it was 42 s) – compared to the earlier detour publications (e.g. Pongrácz et al., 2001, 2003, 2004, 2008) dogs with similar Trial 1 latencies could improve their detour speed significantly after observing human demonstration.
Some minor comments:
L 84-88 considering their relevance, it would be enriching if you briefly describe those findings.
RESPONSE: Thank you for the suggestion, we have elaborated this section, adding these details: “According to the results of the aforementioned studies, high-ranking dogs from multi-dog households show better social learning performance than lower ranking individuals in case of a human demonstrator; however, lower ranking dogs learn much better from a dog demonstrator than high ranking dogs do [32]. Additionally, those dogs that behave assertively and show stronger tendency to be aggressive with their owner, performed better in the detour task where an unfamiliar experimenter demonstrated the task [33]. Finally, we have found that the personality traits that are associated with being more dominant in the hierarchy, could also enhance dogs’ responsiveness to social stimuli in a learning context [34].”
L 148 as you stated the two possible hypotheses; you might clarify the exploratory nature of the study
RESPONSE: Thank you for the comment, indeed, we gave equal opportunity for both hypotheses. Therefore the word “exploratory” was added as you recommended. The text now reads: “In our exploratory study, we used the well-known detour paradigm…”
L 178 add maximum
RESPONSE: Maximum age was added, now the text reads: “We tested adult companion dogs (minimum age 1 year, maximum age 12 years, average ± SD= 5.03 ± 3.08 years).”
I think that it is important to analyze possible effects of sex, age and neutered status.
RESPONSE: Thank you for the recommendation. We have added the sex of dogs among the independent factors (along with training level and keeping conditions). We re-run the Cox models with sex in case of the Trial 1-2-3 latencies both in case of reaching the reward and turning at the end of the fence. We added the results to Table 3. Sex had no significant effect in any of these comparisons. Neuter status was not added as a factor because almost all subjects were neutered/spayed. Regarding dogs’ age, earlier we found (Pongrácz et al., 2005) that among adult dogs, there was no effect of age on detour latencies either with, or without demonstration.
Please, include in Figure 1 the position of the cameras.
RESPONSE: We replaced the Figure 1 with a new one, where we added the two cameras, thank you for the request.
It is necessary to mention the ethics considerations.
RESPONSE: We fully agree. We placed the Informed consent statement and the Institutional review board statement to the end of the manuscript as the Journal’s formatting requirements requested.
Did a second experimenter hold the leash?
RESPONSE: Only one experimenter and the owner participated in the test. If it was necessary, while the owner completed the consent form, the experimenter held the leash of the dog.
How long was the familiarization time?
RESPONSE: This period lasted till the experimenter explained to the owner the task and the rules. It took approximately 5 min. We added this detail to the manuscript, which now reads “Upon their arrival, dog owners (O) gave their written informed consent that they were told about the circumstances and general aims of the study. They entered the testing area accompanied by the experimenter (always the same, young woman, P.D.). The experimenter (E) explained to the O what to do, and what not to do, during the test. The dog was allowed to walk around the area on leash, however, we did not let the dog go behind the V-shaped fence at this time. This initial ‘familiarization’ period took approximately 5 minutes.”
How many owners did not follow the instructions?
RESPONSE: Fortunately, we did not have to exclude any dog because the owner did not follow the instructions regarding how to behave during the test. There was one dog that had to be excluded as it got loose (pulled the leash out from the hand of its owner) while the demonstrator performed the detour. This dog is mentioned in section 2.5 (Exclusions).
Please, clarify if dogs received any reward when they went back to the starting point during failed trials. It is important in order to evaluate the loss of motivation throughout trials.
RESPONSE: We specifically avoided provisioning the dogs with additional treats during the experiment. This was done similarly in all the previous detour experiments. As the experiment consists of only three short trials, being initially unsuccessful (let’s say, in Trial 1), would not cause deleterious loss of motivation. However, if dogs would receive treats after being unable to get the food from behind the fence, they could learn quickly that not performing the detour still yields in reward. In the experiment, we had many subjects who started with 60s (unsuccessful) trial(s), but even missing one or two rewards, they still managed to detour in the subsequent trial(s).
In the demonstration groups it would be clearer if you describe the first trial as a baseline.
RESPONSE: We added the term ‘baseline’ to Trial 1 of the demo-groups at instances where it was appropriate throughout the methods, results and discussion.
Which was the inter-trial interval?
RESPONSE: Inter-trial interval was around 1 minute, we added this to the manuscript now. It reads: “The dog had 60s to solve the task – if it performed a successful detour within the time limit and obtained the reward from behind the fence, the O had to recall the dog to the starting point, and the next trial started. If the dog did not perform a successful detour in 60s, the trial ended and the O had to bring the dog back to the starting point. Two consecutive trials were separated by approximately 1 min inter-trial intervals.”
The colors of the figures are difficult to see, I suggest using darker ones.
RESPONSE: We replaced Figures 2a-2b and Figures 3a-3b with nicer ones.
It would be more informative if you show the performance of each group in Figures 4 and 5.
RESPONSE: Thank you for the suggestion – however, we would prefer keeping the original Figures 4 and 5 because they mirror better the only significant effect (repeated trials), while breed group and testing condition did not have a significant effect on these variables (frequency of owner encouragement and frequency of looking back).
Based on your conclusion that independent breeds detoured the fence with similar latencies compared to the cooperative dogs in the first trial but they did not improve their detour speed after observing the demonstrator, I would expect differences between them in the third trial.
RESPONSE: We did not find group-related difference neither in Trial 2 or Trial 3. The significant effect emerged in case of within-group comparisons – between the baseline (Trial 1) and subsequent Trial 2-3 latencies of the cooperative dogs with human demonstration.
L 549 May be a useful measure would be the frequency of approaching the vertex first.
RESPONSE: This is a very interesting idea, thank you. We will definitely measure this sin our next experiment. Just intuitively answering now, almost each dog approaches the vertex first, especially in Trial 1. Those who detour the fence successfully though, quickly passes by and runs towards the rear end. It is rather rare that dog totally avoid the vertex and go to the rear end first, as the vertex is the closest part of the fence to the starting point.
L 598, please, replace the word “suspect”.
RESPONSE: We replaced the word “suspect” with “speculate”. Now the text reads “We speculate that they might be more sensitive to ostensive communication than independent dogs,…”
Reviewer 2 Report
I congratulate the authors on this highly relevant research. The manuscript is well written and easy to understand. There are only a few minor flaw that I would like to point out:
1. Please check the parentheses through out the manuscript.
2. Please improve the figures (see comments in text).

There are only a few minor grammatical errors (e.g., a dogs').
Author Response
RESPONSES to Reviewer 2
We thank to Reviewer 2 for his/her kindness. We really appreciate it. We also appreciate the thoughtful questions that Reviewer 2 placed into the annotated manuscript. We answer them point by point here.
I congratulate the authors on this highly relevant research. The manuscript is well written and easy to understand. There are only a few minor flaw that I would like to point out:
- Please check the parentheses through out the manuscript.
RESPONSE: We did our best to add missing parentheses wherever we found one missing.
 Please improve the figures (see comments in text).
RESPONSE: We swapped Figures 2a-2b and Figures 3a-3b to better quality ones.
Comment about specific working line/show line dogs in our sample (Table 1)
RESPONSE: Unfortunately, we only asked the owners about the level of training of their dogs. There were some highly trained working dogs in both breed groups, however, we do not have the knowledge about our subjects’ lineage. This is an interesting question though, we agree. The participating people with their dogs were mostly ordinary companion dog owners and only a few of them reported high level sports activities with their dogs.
Comment about the fenced area where the tests were conducted:
RESPONSE: We rewrote the particular sentence that the Reviewer found as being odd. This reads now as: “The fenced-in area provided a calm undisturbed environment where no other dogs or bystanders could interrupt the procedure.”
Comment about ‘biddability’
RESPONSE: We added this short description to the term “biddability”: “responsiveness to human direction and commands”
Reviewer 3 Report
Dear authors,
thanks very much for this excellent paper. I have just one short remark.
I can follow your selection process and decisions on your test group. And the results speak in favor for your hypothesis 1. Still I would like a sentence in the discussion/summary that your work provides relevant information, as seeing the improvement from trial 1 to trial 2/3 in the dogs from the demonstration group speaks for itself. But as many factors might influence socio-cognitive behavior (e.g. not only whether the dog underwent some training but, maybe more important, how this training was conducted), more such experiments, and with dogs not only from the working dog area, need to be conducted.
Author Response
RESPONSES to Reviewer 3
We are thankful for your supportive comments and kind attitude. It means a lot!
Dear authors,
thanks very much for this excellent paper. I have just one short remark.
I can follow your selection process and decisions on your test group. And the results speak in favor for your hypothesis 1. Still I would like a sentence in the discussion/summary that your work provides relevant information, as seeing the improvement from trial 1 to trial 2/3 in the dogs from the demonstration group speaks for itself. But as many factors might influence socio-cognitive behavior (e.g. not only whether the dog underwent some training but, maybe more important, how this training was conducted), more such experiments, and with dogs not only from the working dog area, need to be conducted.
RESPONSE: Very interesting thought. We think it is important to state that none of the subjects had any previous experience (training) with similar detour tests. This actually was one of the few requirements for participating in this particular experiment. Therefore, we added this text to the ‘Subjects’ section:
“We specified which dog breeds we were looking for, as well as the minimum age limit. We also required that the subjects had not previously participated in a detour test. Apart from these, we did not have any further requirements.”
We also thought it would be interesting to mention in the conclusions that the so-called show-lines and working-lines that exist in particular dog breeds, could perform differently in a test like the one we run here. Also, we appreciated your suggestion about the potential effect of how and for which purpose dogs are trained. We wrote the following paragraph at the end of the Discussion:
“It is worth mentioning that particular dog breeds comprise of already distinct ‘working’ and ‘show’ lines, which are under different pressures of artificial selection. In the future, it would be interesting to see whether dogs from these lines would perform differently in such experiments that we described here. Also, although we found no significant effect of the dogs’ training levels, one could hypothesize that particular forms or types of training could still influence the trial-and-error or social learning capacity of dogs. Time and further experiments can only tell whether such factors would be relevant in case of dogs’ social learning from humans.”